# Nitrogen-Doped Zinc Oxide for Photo-Driven Molecular Hydrogen Production

**DOI:** 10.3390/ijms23095222

**Published:** 2022-05-07

**Authors:** Erik Cerrato, Alberto Privitera, Mario Chiesa, Enrico Salvadori, Maria Cristina Paganini

**Affiliations:** 1INRIM Istituto Nazionale di Ricerca Metrologica, 10135 Torino, Italy; e.cerrato@inrim.it; 2Department of Chemistry and NIS Centre, University of Turin, 10125 Torino, Italy; alberto.privitera@unito.it (A.P.); mario.chiesa@unito.it (M.C.)

**Keywords:** semiconductors, zinc oxide, nitrogen doping, photocatalysis, EPR spectroscopy

## Abstract

Due to its thermal stability, conductivity, high exciton binding energy and high electron mobility, zinc oxide is one of the most studied semiconductors in the field of photocatalysis. However, the wide bandgap requires the use of UV photons to harness its potential. A convenient way to appease such a limitation is the doping of the lattice with foreign atoms which, in turn, introduce localized states (defects) within the bandgap. Such localized states make the material optically active in the visible range and reduce the energy required to initiate photo-driven charge separation events. In this work, we employed a green synthetic procedure to achieve a high level of doping and have demonstrated how the thermal treatment during synthesis is crucial to select specific the microscopic (molecular) nature of the defect and, ultimately, the type of chemistry (reduction versus oxidation) that the material is able to perform. We found that low-temperature treatments produce material with higher efficiency in the water photosplitting reaction. This constitutes a further step in the establishment of N-doped ZnO as a photocatalyst for artificial photosynthesis.

## 1. Introduction

Heterogeneous photocatalysis is a research area that has received a great deal of interest in recent decades. The main goal of photocatalytic processes is the exploitation of energy associated with light photons to promote chemical transformations. The photocatalytic approach is general and can be pursued in all types of chemical processes in which the required activation energy can be provided by electromagnetic energy. The most popular applications include pollution remediation (total oxidation of pollutants), CO_2_ reduction and solar fuel production. The latter is often based on the water photosplitting reaction for H_2_ production [1,2] and is related to field of artificial photosynthesis [3]. However, while laboratory set-ups have shown promising results with tailored irradiation systems such as lamps and lasers, the technological application of photocatalytic processes to larger scales will require the efficient use of sunlight. In fact, the electromagnetic radiation emanated by the Sun that reaches the Earth’s surface spans the region from the ultraviolet to the infrared, but has a prominent peak in the visible range, roughly between 400 and 700 nm.

Metal-oxide semiconductors with high band gaps, such as titanium dioxide (TiO_2_, 3.2 eV) or zinc oxide (ZnO, 3.2–3.4 eV), have so far dominated the field of environmental photocatalysis. These have outstanding photocatalytic properties and show remarkable photostability, but require the use of UV light to function as photocatalysts. Moreover, while the photoexcited holes in the valence band (VB) show optimal oxidation potential that can be conveniently used, for instance for the mineralization of pollutants, the photoexcited electrons in the conduction band (CB) have an inefficient reductive potential for the water photosplitting reaction used to produce molecular hydrogen. In order to enlarge the photon response of the materials and to increase the electron reductive potential, a wide array of strategies have been developed and improved during the years and all resulted in photocatalysts that are able to carry out both reduction and oxidation reactions. Examples include the sensitization of the material by coating with organic dyes working as “antenna” for visible solar photons, contacting two semiconductors to form a heterojunction or the introduction of foreign atoms inside the oxide lattice [4,5,6,7,8]. All of these strategies result in photocatalysts that are able to carry out both reduction and oxidation reactions.

In particular, the latter case of semiconducting systems with band gaps modified by the insertion of impurities is well represented by N-doped TiO_2_, a system that has served as an excellent testbed for fundamental studies [4,9,10].

From such studies, it has become apparent that the electronic and magnetic properties displayed by a semiconductor in the presence of a certain concentration of dopants or defects, which are often associated to trapping or self-trapping phenomena, are crucial in determining the performance of these materials in the various applications. Nitrogen has emerged as an important doping element in different semiconductors oxides. Using N-doped TiO_2_, it has been largely demonstrated that the N 2p impurity states are localized on the N atom and their energy levels lie a few tenths of an electronvolt above the VB [11,12].

As far as zinc oxide is concerned, the incorporation of N atoms in the ZnO lattice is a very stimulating challenge considering that it is well known that ZnO is an intrinsic *n-type* semiconductor (excess of electrons) and the insertion of N (one less electron than O) in the lattice of ZnO has been proposed as an effective way to achieve *p-type* doping (excess of holes) of this material. Achieving *p-type* doping is fundamental for optoelectronic as well as photochemical applications [13,14]. However, the intrinsic *n-type* nature of ZnO (which sets apart this material from TiO_2_) represents a considerable hindrance to the effective stabilization of impurities acting as shallow acceptors. The main difficulties have been reported to originate from the compensation by low-energy native defects. As an example, Iwata et al. did not observe a ZnO *n-type* to *p-type* conversion adopting molecular-beam epitaxy (MBE) as a synthetic route; rather, the simultaneous introduction of O_2_ and N_2_ via a RF radical source just resulted in an increase in the intrinsic point defects and in the instauration of deep states concerning the nitrogen impurity [15].

Previous research has demonstrated that N atoms can be incorporated in the host ZnO crystal lattice by annealing polycrystalline ZnO in an NH_3_ atmosphere [16,17]; however, this gas-based method is difficult to implement experimentally and only yields a low level of doping. Alternative strategies entailed the use of N_2_, NO, and N_2_O as nitrogen sources, however they led to both poor N incorporation in the ZnO matrix and a low concentration of shallow donors [5,18,19,20,21]. An exception is represented by Refs. [22,23], where the authors reported an enhancement of N doping by thermal annealing in N_2_ at high temperatures (600–900 °C). The presence of shallow acceptors, attributable to substitutional nitrogen, was observed through photoinduced electron paramagnetic resonance (EPR) spectroscopy.

Finally, despite its potential activity, especially when effective shallow donors due to nitrogen insertion were recorded, N-doped ZnO materials have been poorly tested for photocatalytic applications upon visible irradiation (in particular concerning H_2_ production via the water photosplitting process) [24,25,26,27,28,29,30,31]. These considerations prompted the design of the experiments reported in this paper.

In this work, an easy and facile synthesis has been successfully adopted to introduce nitrogen in the ZnO lattice via a wet chemistry procedure, starting from a cheap and environmentally friendly precursor, (NH_4_)_2_CO_3_, as the source of (reduced) nitrogen. The co-precipitation method assures that a high level of doping can be achieved. The data we present demonstrate how the calcination temperature plays a crucial role in determining the nature and concentration of the nitrogen species trapped in the lattice and the overall chemistry displayed by the material. On the basis of previous literature [16,22,23], the spectroscopic signatures of monomeric (substitutional N^•2−^) and dimeric (N^•^_2_^−^) nitrogen-based impurities were identified. The former dominates in the material treaded at 500 °C, whereas the latter dominates in the material treated at 300 °C. Most importantly, the calcination temperature affects the type of chemistry N-doped ZnO is able to perform—N-doped ZnO calcinated at 500 °C (hereafter NZnO_500) is more efficient in photo-driven oxidative processes (e.g., the generation of reactive oxygen species such as OH^•^ radicals), while N-doped ZnO calcinated at 300 °C (henceforth NZnO_300) is more effective in photo-driven reductive chemistry, as shown by the higher production yield of molecular hydrogen following the water photosplitting reaction.

## 2. Results

### 2.1. XRD Characterization

Each sample was analyzed from a structural point of view by means of X-ray powder diffraction. The collected XRD diffractograms are presented in Figure 1 and show the typical pattern of ZnO wurtzitic hexagonal phase (00-036-1451 ICDD pattern) [32]. The reported patterns indicate the absence of secondary phases in the synthetized samples and allow for the evaluation of the degree of crystallinity of the material. In particular, while the diffractograms of ZnO and NZnO_500 show the highest and sharpest reflections and are virtually indistinguishable, the diffractogram of NZnO_300 reveals broader peaks. In turn, this indicates higher crystallinity and larger crystallite size for the former and smaller crystals for the latter. These qualitative conclusions were also confirmed by the application of the Scherrer equation [33] to the XRD patterns of Figure 1, revealing that ZnO and NZnO_500 have bigger crystallite sizes, with an average size of 338 ± 21 nm, while NZnO_300 presents a smaller crystallinity and an average size of 85 ± 10 nm.

Complementary XPS experiments on NZnO_300 and NZnO_500 (not shown) revealed nitrogen and carbon contents of ≈1% and ≈10%, respectively, for both samples. In turn this implies that the characteristic chemistry displayed by each material (vide infra) is solely due to the nature of the defective species formed and not to the nitrogen content. The origin of the carbon signal may be due to adventitious carbon as well as CO_2_ incorporated during synthesis, but this has no effect on the properties of the materials (vide infra).

### 2.2. UV-Vis Diffuse Reflectance

Figure 2 reports the normalized absorbance spectra. All the samples are dominated by the valence band—a conduction band direct electronic transition typical of pristine ZnO [34]. The energy gap of the synthesized materials was evaluated using the Tauc equation [35] and found to be in agreement with the value of 3.3 eV, which is commonly reported in the literature [36]. However, it is worth nothing that, as compared to ZnO, the optical transitions of both NZnO samples are less sharp and a broad absorption shoulder can be observed between 450 nm and 600 nm, see Figure 2. While both nitrogen-doped materials can be visually discriminated from pristine ZnO given that they present a pink/orange coloration, the insets of Figure 2 show that the visible absorption band of NznO_300 is ca. 2.5 times more intense than the corresponding band of NznO_500.

The remarkable conclusion that can be drawn from inspection of Figure 2 is that the absorption band for the nitrogen-doped materials is wide and broad between 400 and 600 nm, making this material capable of harvesting even the yellow-red region of the solar electromagnetic spectrum. Moreover, it is evident that the sample NZnO_300 presents an absorption band with a maximum at a lower wavelength with respect to the sample NZnO_500; this behavior will be explained later with the identification of two different nitrogen species generated in these samples. Nitrogen doping, and hence the creation of defective levels within the bandgap, is a methodology that has been used to sensitize oxides to visible light absorption. From the spectra reported in Figure 2, it can be proposed that the visible band is either related to the excitation of electrons from the oxide valence band to the defective intraband gap states or from the latter to the conduction band of the material.

### 2.3. EPR Spectroscopy

In order to provide a description—at the molecular level—of which defects are formed by nitrogen insertion in the ZnO lattice, electron paramagnetic resonance (EPR) spectroscopy was used. EPR spectroscopy was coupled with in situ irradiation to directly monitor the photoactivity of the materials and to establish where the photoinduced charge carriers stabilize [37,38]. In this respect, it should be considered that, when the irradiation is performed at low temperature (<77 K) and in vacuum, photogenerated electrons and holes may stabilize at specific locations in the solid. Given that both electron and holes have spin angular momentum (S = 1/2), they can be detected through EPR spectroscopy.

The low temperature (80 K and 10 K) EPR spectra of the materials studied are reported in Figure 3 in dark and after monochromatic laser irradiations at different wavelengths in the visible range.

Firstly, it is interesting to evaluate the nature of the defects present in the material spectra before the laser irradiation (black lines) in order to identify the species responsible for the photoactivity in the visible. It is worth recalling that NZnO is known to display three families of EPR signals stemming from intrinsic defects and defects formed during nitrogen incorporation and thermal annealing—these have been previously reported and discussed [16,39]. The signal at g_av_ = 1.959, always present in ZnO, with and without nitrogen doping, has been widely reported and attributed to shallow donor impurities a few meV below the conduction band [40,41,42,43,44,45]. The signal is usually symmetric; however, it may be split at high field if bulk and surface sites are present at the same time. A second species, clearly visible in the spectra of NZnO_300 at 10 K and NZnO_500 (both 10 K and 80 K), can be identified by an axial g matrix [g_x_, g_y_, g_z_] = [1.9630, 1.9630, 1.9947] (g_av_ = 1.974) and a characteristic hyperfine coupling with a single nitrogen nucleus [A_x_, A_y_, A_z_] = [8.6, 8.6, 81.3] MHz. This species may be described in molecular terms as a N^•2−^ ion with electronic configuration 1s^2^2s^2^2p^5^, where the unpaired electron is localized in one of the nitrogen 2p orbitals [16,17,22,46,47]. Such a configuration can be alternatively pictured as a positive hole localized on a nitride (N^3−^) ion and represent a substitutional nitrogen (N replacing O) in the lattice. Analysis of the hyperfine parameter yielded a total spin density at the nitrogen nucleus of ≈0.7, demonstrating that the spin is strongly localized on the N nucleus. The third species can be described by an axial g matrix [g_x_, g_y_, g_z_] = [1.9935, 1.99935, 2.0037] (g_av_ = 1.997). The number of hyperfine lines clearly distinguishable in the spectrum imply that the unpaired electron is coupled to two magnetically equivalent nitrogen nuclei with hyperfine parameters [A_x_, A_y_, A_z_] = [20.1, 20.1, 10.1] MHz. This species is only visible in the EPR spectrum of NZnO_300 at 80 K, while it is not present for NZnO_500 at either 10 or 80 K.

Such a signal has been attributed to a molecular N_2_^●−^ center [23,48,49]; in this case, the unpaired electron occupies the π_x_* antibonding orbital with the degeneracy of the two π_x_* and π_y_* orbitals removed by asymmetric perturbation of the surrounding Zn ions. The reduction in symmetry indicates a strong interaction of the two species [49]. Interestingly, the annealing temperature used in this study affects the type of nitrogen defect present in the final products, as demonstrated by the comparison of the EPR spectra recorded at 10 K and 80 K for the two samples. At 10 K, NZnO_300 displays the g_av_ = 1.959 EPR signal typical of shallow donors. A minor contribution from the substitutional N^2−^ ion is visible (for instance, at 345–355 mT). On the other hand, at 10 K, NZnO_500 only displays a well-structured and intense signal typical of an N^•2−^ signal. This is consistent with the lower nitrogen content in the sample due to the higher annealing temperature, with a similar results reported in [22,23]. In particular, the authors annealed the N-doped ZnO (produced via chemical vapor transport method) at specific high temperatures in static air, identifying a turning point around 500 °C where the amount of N^•2−^ becomes dominant with respect to N_2_^●−^. The authors justified this trend by considering the formation of nitrogen-hydrogen complexes: the dissociation of these neutral complexes (EPR silent) would be activated by annealing temperature higher than 450 °C, giving rise to a higher amount of singly ionized nitrogen species N^•2−^. These would dominate the spectrum of the NZnO_500 sample, masking the signal of the molecular species N_2_^●−^, as shown by the spectra reported in Figure 3b.

The observation of N_2_^●−^ at 80 K is at odds with most previous literature that reported fast relaxation times for the N_2_^●−^ center and the need for very low temperature to observe its characteristic EPR signature. However, Garces et al. observed a similar behavior in their N-doped ZnO, the signal of N_2_^●−^, monitored in the temperature range 50–90 K, recovered following light irradiation.

It is worth mentioning that no signals attributable to the presence of NO^•2−^, the main photoactive species in N-doped TiO_2_ (system photoactive in the visible), were identified [4,10,50,51]. This reinforces the notion that the effect of N insertion in the lattice is strongly dependent on the chemical nature of the oxide. This also affects the evolution of the paramagnetic species upon photoexcitation. To this end, we selected a few monochromatic excitation wavelengths largely below the material bandgap (500, 600 and 700 nm), irradiating the samples within the EPR spectrometer at low temperature, and compared the EPR spectra under illumination with a reference spectrum recorded in the dark. NZnO_300 appears to be very sensitive, with both the EPR signals (due to shallow donors and N_2_^●−^ centers) increasing considerably after illumination, regardless of the wavelength used. The signal of N^•2−^ is likely still present but masked under the signal of N_2_^●−^. On the other hand, NZnO_500 displays a more modest response, the signal intensity only marginally increases upon illumination and no other species appear. In summary, EPR experiments suggest that the molecular N_2_^●−^ species are responsible for the higher optical response of NZnO_300 and that the calcination temperature causes loss of nitrogen from the lattice and the conversion of N_2_^●−^ to N^•2−^.

We note that, despite the relatively high carbon concentration detected through XPS, EPR does not reveal any signal ascribable to carbon centered species.

To correlate the microscopic structure of the defects to specific chemical reactions, the materials were tested for both reductive and oxidative photocatalytic reactions promoted by visible frequencies.

### 2.4. Spin Trapping

To gain insight into the photocatalytic mechanisms, specifically the photoactivity at the solid–liquid interface, the materials were tested through spin trapping in solution. An aqueous suspension of each sample was irradiated in the presence of the spin trapping DMPO, which is particularly useful to detect the formation and provide a relative quantification of the hydroxyl radical (OH^●^). The reaction between OH^●^ and DMPO leads to the formation of the persistent radical adduct DMPO-OH, which is characterized by a four-line EPR spectrum with relative intensities 1:2:2:1 [52]. The photo-induced formation of OH^●^ can be accounted for by the reaction of either water molecules or surface hydroxyl groups with photogenerated holes, according to the chemical Equations (1) and (2):

h^●+^_(VB)_ + H_2_O -> OH^●^ + H^+^(1)

h^●+^_(VB)_ + OH^−^ -> OH^●^(2)

Figure 4 compares the intensities of the EPR spectra of DMPO-OH obtained after irradiating each sample (NZnO_300, NZnO_500 and a control ZnO) in the visible for 30 min at 610 nm. By considering the signal intensity of the spectra, it could be concluded that NZnO_500 is five times more effective in producing OH^●^ radicals than NZnO_300. Interestingly, NZnO_300 appears less effective even than the control ZnO. Broadly speaking an efficient yield of photo-induced OH^●^ radicals is of relevance in all those processes that employ photons to initiate degradation events. The higher yield of OH^●^ radicals displayed by NZnO_500 points towards an oxidative chemistry promoted by this material. These data suggest that the acceptor nitrogen species introduced in the material annealed at higher temperature is very active in the production of hydroxyl species, even at wavelengths that are much smaller than the bandgap.

### 2.5. H_2_ Production

In order to explore the full photocatalytic ability of the prepared samples, NZnO_300 and NZnO_500 were tested for the generation of H_2_ via the water photosplitting process under either visible or UV light. The water photosplitting process (Equation (3)) can be depicted as the sum of two half redox reactions: the hydrogen evolution reaction (HER) and oxygen evolution reaction (OER). In the HER, the redox potential of the photogenerated electron in the conduction band is exploited to reduce two protons (H^+^) to a hydrogen molecule (Equation (4)), whereas the redox potential of the hole in the valence band is utilized to oxidize O^2−^ anions to molecular oxygen (Equation (5)). A semiconductor can drive such a reaction only if the highest level of the VB is more positive than the water oxidation reduction potential (1.23 V against the normal hydrogen electrode (NHE)) and the edge of the conduction band is more negative than the hydrogen evolution potential (0 V against NHE). The energy values for the valence band and conduction band in ZnO are +2.89/−0.31 Ev, respectively.

H_2_O -> H_2_ + ½O_2_(3)

2H^+^ + 2e^−^ -> H_2_                        E^0^_red_ = 0.00 V(4)

H_2_O + 2h^+^ -> 2H^+^ + 1/2O_2_     E^0^_ox_ = −1.23 V(5)

The experimental set-up used to assess the yield of the water photosplitting of N-ZnO only monitored H_2_ production (HER, Equation (4)). The number of H_2_ μmol produced was used as a quantitative means to compare the efficiency of the materials. Figure 5 reports the measured H_2_ production after 2 h of the corresponding irradiation. Figure 5 demonstrates that N-ZnO materials are more active in the H_2_ production with respect to pristine ZnO, both under UV as well as visible irradiation. Not surprisingly UV irradiation always corresponds to higher activity given that UV photons have sufficient energy to directly promote an electron from the valence to the conduction band. Interestingly, however, and at odds with the spin trapping experiments that demonstrated a higher activity of NZnO_500 towards oxidative chemistry, NZnO_300 shows a higher photoactivity in the splitting of the water molecule and in the reduction of protons to H_2_.

## 3. Discussion

In light of the spectroscopic and photoactivity data presented in this study, it is possible to elucidate the effect of the calcination temperature on the molecular nature of the defect present in NZnO and on the chemistry that the material is able to drive. EPR spectroscopy indicates that N doping results in the formation of different acceptor species whose relative contribution is a function of the calcination temperature—while NZnO_300 is characterized by the presence of both N^•2−^ (minor component) and N_2_^●−^ (major component) species, NZnO_500 only shows fingerprints ascribable to N^•2−^ centers. Low-temperature irradiation at visible wavelengths much longer than the bandgap (<2.4 eV) also results in distinct behaviors. Illumination of NZnO_300 results in a significant increase of N_2_^●−^ species that make the N^•2−^ contribution almost undetectable. On the other hand, the photoactivity of NZnO_500 is limited, and only a slight increment of the N^•2−^ signal could be detected.

This nature and the number of the defects present in the material as a consequence of the calcination temperature also affect the behavior with respect to the photocatalytic capability at the solid–aqueous interface, both in oxidation and reduction reactions. NZnO_300, which affords the highest concentration of N_2_^●−^ centers, is more efficient in the reduction of H^+^ to H_2_ during water photosplitting, whereas NZnO_500, which is characterized by the highest concentration of N^•2−^ centers, is more effective in the oxidation of water (or hydroxyl ion) to form hydroxyl radicals.

The distinctive chemistry displayed by the two materials must reflect the different populations and nature of defects of species within the bandgap of ZnO. In turn, this only depends on the post-synthesis calcination temperature.

Considering all the experimental data discussed, Figure 6 tentatively models the bandgap diagram for NZnO as a function of the calcination temperature.

The model explains both the UV-visible spectra and the EPR spectra, at the same time it gives a possible interpretation of the photoactivity exhibited by the two samples. Essentially, the band diagram of NZnO_300 appears highly defective, showing both N^•2−^ and N_2_^●−^ centers. These, in turn, correspond to shallow and deep acceptor levels in the ZnO band gap. In the model, all the possible nitrogen species are reported. For the monomer species, N^−^ and N^3−^ are diamagnetic, respectively empty and double occupied, and N^•2−^ is paramagnetic. For the dimeric species, N_2_^0^ and N_2_^•2−^ are diamagnetic and N_2_^●−^ is paramagnetic. This is somehow supported by the two different absorption shoulders visible in the UV-vis spectrum. The high concentration of N_2_^●−^ defects in the NZnO_300 sample may create deep levels in the band gap that can be populated under irradiation with a red light (610 nm/2.4 eV). According to our previous experience with N-doped TiO_2_ [10], it is possible to hypothesize also in this case a double step of irradiation. The existence of the nitrogen intraband gap allows the system to draw the needed quantum energy for moving electrons from VB to CB via two distinct steps rather than from a direct single step. The presence of these localized states allows a double excitation process. Moreover, this might also explain the lower ability in photo-oxidative reactions. In fact, the hole stabilized at a N defect center may have a higher oxidative potential than the H_2_O/OH^•^ couple, thus preventing hydroxyl formation. In any case, the data presented here do not allow for the derivation of any definitive conclusions and this point still remains an open question and further specific spectroscopic experiments involving pulse-EPR, as well as accurate theoretical calculations, will be needed to shed the light on this crucial point. On the other hand, NZnO_500 shows only N^•2−^ defects behaving as shallow acceptors, located a couple of meV above the valence band edge, as reported in the literature. Such species play a similar role as NO impurities in N-doped TiO_2_ [4,10,51]. These shallow acceptors are responsible for the high production of holes and for the oxidative chemistry displayed by NZnO_500.

## 4. Conclusions

In this work, we have used complementary spectroscopic techniques to assess the effect of the calcination temperature (thermal treatment) in the formation of specific types of defects within the crystal lattice of ZnO following nitrogen doping. To achieve a high level of doping and avoid the use of toxic ammonia, nitrogen was introduced via wet chemistry methods, using (NH_4_)_2_CO_3_ as nitrogen precursor. Through the combination of spectroscopic characterization and model chemical reactions, we have demonstrated that the thermal treatment affects the nature of the defects formed, which in turn determines the optical response of the material and the chemistry that it is able to drive. Specifically, high-temperature treatment favors oxidative chemistry, while low-temperature treatment drives reductive chemistry. This work extends the understanding of N-doped ZnO in light of applications for artificial photosynthesis.

## 5. Materials and Methods

All reactants used in the synthetic procedures were acquired by Sigma-Aldrich (Merck, Darmstadt, Germany) and used without any additional purification treatment. Distilled water was used in all synthetic procedures. The investigated samples were mainly prepared via a facile precipitation synthetic route, described hereafter.

Synthesis: ZnO nanoparticles were synthetized via a precipitation method either in the presence or absence of (NH_4_)_2_CO_3_ and subsequently calcinated. In this way, three samples were produced that hereafter will be referred to as ZnO (pristine zinc oxide, no nitrogen doping, thermally treated at 300 °C), NZnO_300 (nitrogen doped zinc oxide thermally treated at 300 °C) and NZnO_500 (nitrogen doped zinc oxide thermally treated at 500 °C).

### 5.1. N-Doped ZnO Synthesis

The N-doped ZnO samples were synthesized by adding 6.72 g of (NH_4_)_2_CO_3_ in 20 mL of H_2_O (solution A), thus generating an oversaturated solution with a basic pH around 8–9. Solution B was prepared by dissolving 5.95 g of Zn(NO_3_)_2_ in 20 mL H_2_O. Once the zinc nitrate was completely dissolved in water, solution B was added dropwise to solution A. The obtained white precipitate was left to rest at room temperature for 4 h and then dried in the oven at 70 °C overnight. The obtained white powder was then divided in to two batches—one was calcined in air for 3 h (ramp of 5 °C/min) at 300 °C, while the second at 500 °C.

### 5.2. Powder XRD Diffraction and X-ray Photoemission Spectroscopy (XPS)

The crystal phase of the synthetized materials was verified with X-ray powder diffraction (XRPD). The diffraction patterns were recorded with a PANalytical PW3040/60 X’Pert PRO MPD (Malvern Panalytical Ltd, Malvern, United Kingdom) diffractometer using a copper Kα radiation source (0.154056 nm). The XRD patterns were obtained in the range 20° < 2θ < 80°. X’Pert High-Source software (Malvern Panalytical Ltd, Malvern, United Kingdom) was used for data handling.

X-ray photoemission spectroscopy (XPS) measurements were used to quantify the amount of nitrogen incorporated during synthesis. Spectra were collected in the main chamber of the ultra-high-vacuum system at a base pressure of 5 × 10^–9^ mbar at room temperature, equipped with an Omicron DAR 400 X-ray source (Al Kα = 1486.7 eV) and a VG Mk II Escalab electron analyzer. As a result of homogeneous charging, the binding energy (BE) scale was shifted using the adventitious carbon signal and setting its position to 284.8 eV. The photoemission lines were analyzed using the XPSPEAK 4.1 software, which uses a mixed linear and Shirley background and symmetrical Gaussian-Lorentzian functions.

### 5.3. UV-Vis Spectroscopy

The DR-UV-Vis absorption spectra were registered with a Varian Cary 5 spectrometer (Agilent, Santa Clara, CA, USA), coupled with an integration sphere for diffuse reflectance (DR). A sample of PTFE with 100% reflectance was employed as reference. Spectra were registered in the 200–800 nm range at a scan rate of 240 nm/min with a step size of 1 nm. The measured intensities were converted with the Kubelka–Munk function. The energy gap has been evaluated using the Tauc plot method [35]. The raw data were analyzed with a Carywin UV/scan software (Agilent, Santa Clara, CA, USA).

### 5.4. EPR Spectroscopy

CW EPR experiments were performed on an ELEXYS 580 Bruker spectrometer (Ettlingen, Germany) operating at 9.76 GHz, equipped with a Bruker ER 4118X-MD5 resonator housed in liquid helium cryostat from Oxford Inc (Abingdon, Oxfordshire, United Kingdom). The magnetic field was measured with a Bruker ER035M NMR gaussmeter. The spectra were recorded at 10 K and 80 K. The photo-activity of the synthetized material was investigated by means of coupled EPR spectroscopy with in situ irradiation using a Litron Aurora II OPO Laser (Rugby, Warwickshire, United Kingdom). The sample was irradiated for 2 min. EPR spectra were simulated using the Easyspin [53] software using the function “pepper” and the parameters reported in the main text.

### 5.5. Spin Trapping

Experiments to detect the formation of the OH^●^ radical upon irradiation were performed on a Miniscope 100 spectrometer from Magnettech (Bruker, Ettlingen, Germany). DMPO (5,5-dymethil-1-pyrroline-N-oxide) was used as the spin trapping agent.

### 5.6. Photocatalytic H_2_ Evolution

Molecular hydrogen evolution was tested via the water photo-splitting reaction. Firstly, 1 g of photocatalyst powder was suspended in 20 mL of a 10% *v*/*v* ethanol aqueous solution and sonicated for 10 min in a 100 mL quartz flask reactor, isolated from the external environment. The sample was irradiated with a 500 W Xe lamp with an irradiance of 80 W/m^2^ (measured without filters). A magnetic stirrer was placed at the bottom of the reactor to keep the particles in suspension during the experiment. Prior to illumination, N_2_ was purged into the reactor for 15 min to remove atmospheric oxygen. The photocatalytic activity of the NZnO_300 and NZnO_500 materials was then monitored by illuminating the suspension for 2 h. To quantitatively evaluate hydrogen generation, a Micro Gas Chromatograph GC490 (Agilent, Santa Clara, CA, USA) equipped with a MS5A column using Ar as carrier gas was employed. A 50 ppm H_2_/O_2_ in Ar gas mixture purchased by Savio was used as a standard for quantification.

## Figures and Tables

**Figure 1 ijms-23-05222-f001:**
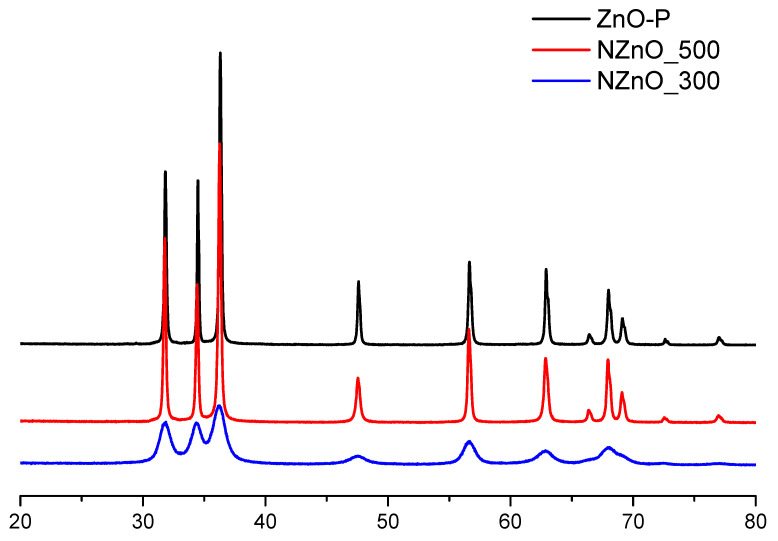
XRD patterns of ZnO (black), NZnO calcined at 500 °C (red), and NZnO calcined at 300 °C (blue) samples.

**Figure 2 ijms-23-05222-f002:**
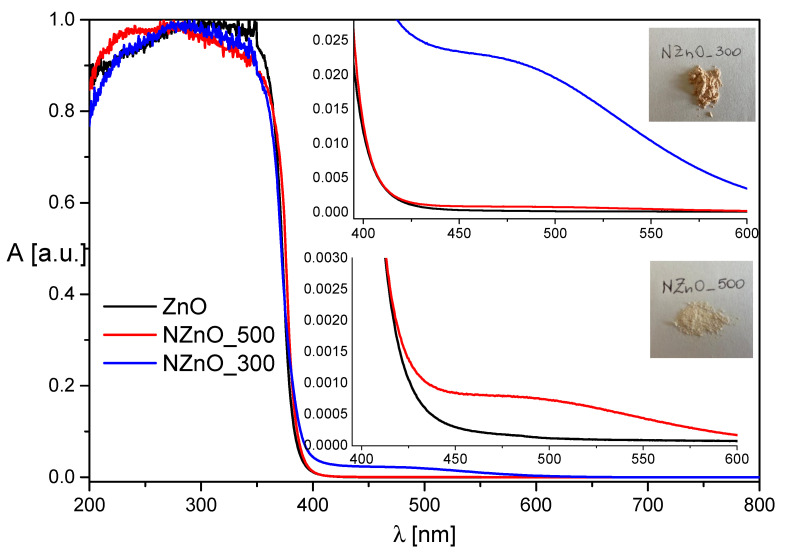
DR-UV-Vis spectra (Kubelka-Munk) of ZnO (black); NZnO_500 (red), NZnO_300 °C (blue) samples. The y-axis reports the absorbance (A) in arbitrary units.

**Figure 3 ijms-23-05222-f003:**
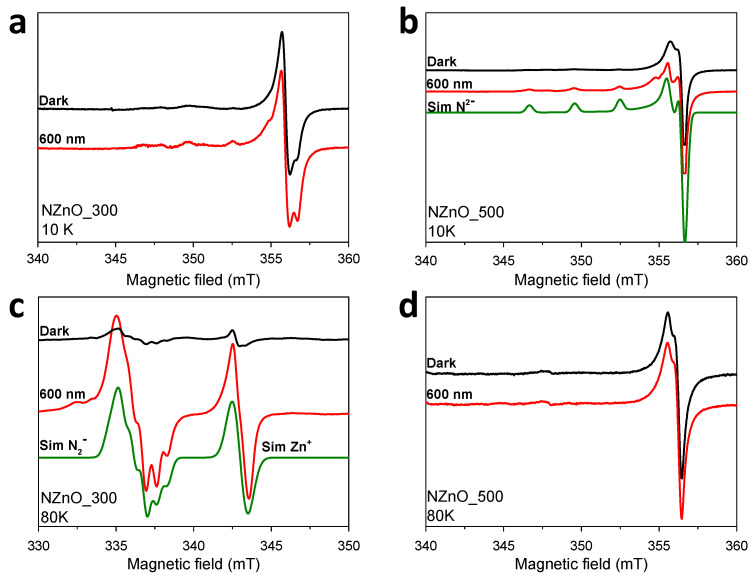
EPR spectra recorded at 10 K and 80 K of NZnO_300 (panel (**a**,**c**), respectively) and NZnO_500 (panel (**b**,**d**), respectively). Black line: before laser irradiation. Red line: after irradiation at 600 nm. The green lines in panels (**b**,**c**) are the simulations for the major components obtained with the parameters reported in the text.

**Figure 4 ijms-23-05222-f004:**
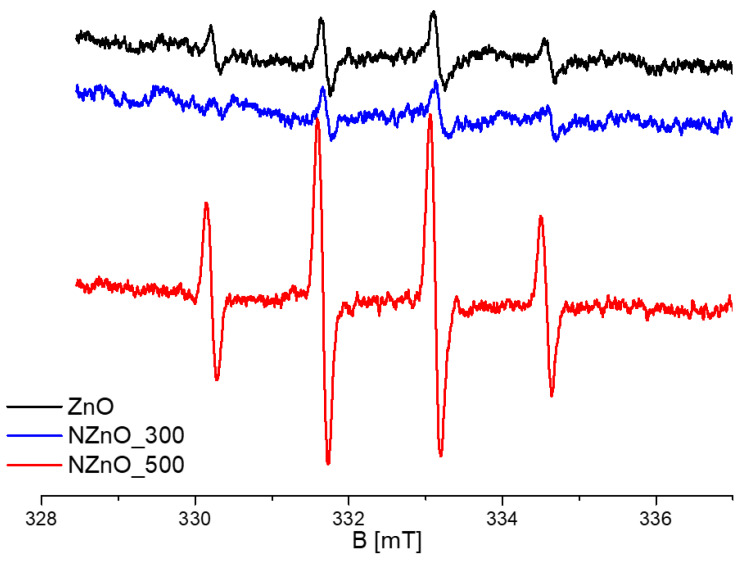
EPR spectra of the DMPO/OH^•^ adduct produced by irradiation of aqueous suspensions of ZnO NZnO_300 and NZnO_500 with visible light at 610 nm. The x-axis represents the applied magnetic field (B) in units of milli-tesla (mT).

**Figure 5 ijms-23-05222-f005:**
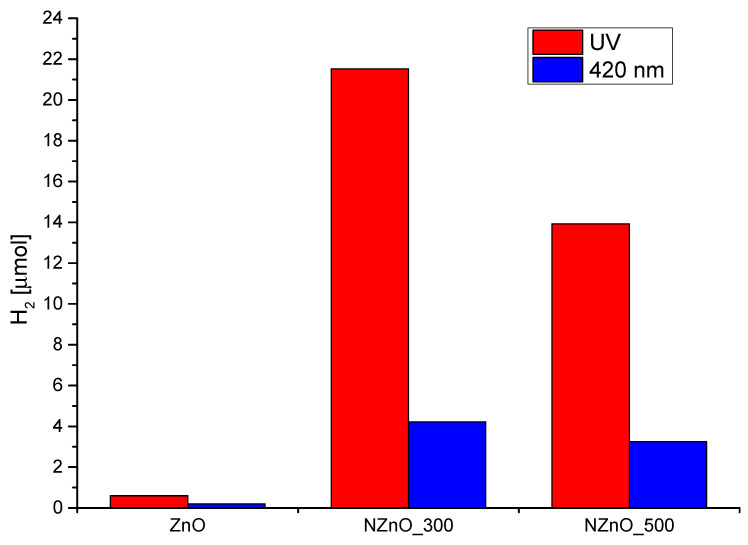
Comparison of H_2_ production under both UV-Vis and visible light for the Zn-based samples.

**Figure 6 ijms-23-05222-f006:**
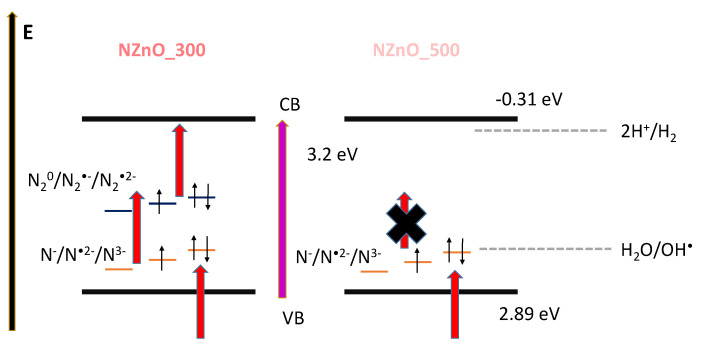
Proposed band diagram for the synthetized NZnO_300 and NZnO_500, respectively. The y-axis represents energy (E). The thick black lines represent the energy position of the valence (VB) and conduction (CB) bands of ZnO. The purple arrow represents the bandgap absorption of 3.2 eV. The dashed lines represent the energy position of the processes described by Equations (1) and (4), respectively. The orange and blue horizontal lines represent the two families of defects present in the materials, while the thin black arrows depict their oxidation and spin states. The red thick arrows represent all the absorption processes involving defect states discussed in the manuscript. The thick black cross on the right-hand side symbolizes a process not active in NZnO_500.

## Data Availability

Not applicable.

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
