# Peer review of "Nitrogen-Doped Zinc Oxide for Photo-Driven Molecular Hydrogen Production"

_ijms, 2022, doi:10.3390/ijms23095222_

Round 1

Reviewer 1 Report

Nitrogen-doped zinc oxide is not a new concept. Synthesis of doped  ZnO from literature has to be given either introduction or in the experimental so that the contribution of the authors can be recognized. 

Figure quality needs to be improved. 

Author Response

Nitrogen-doped zinc oxide is not a new concept. Synthesis of doped ZnO from literature has to be given either introduction or in the experimental so that the contribution of the authors can be recognized. 

We thank the Reviewer for the comment that can help improve the manuscript impact. To this end, we added two paragraphs in the introduction section concerning the main synthetic routes and corresponding outcomes proposed in the past for N-doped ZnO.All the added paragraph are underlined in yellow color

Figure quality needs to be improved. 

We have improved the quality of the figures

Reviewer 2 Report

The paper is well presented, and recommend for publishing.

Author Response

The paper is well presented, and recommend for publishing.

We really appreciated the comment of the referee

Reviewer 3 Report

The authors prepared the N-doped ZnO by the precipitation synthetic method and demonstrated that the thermal treatment during the fabrication process exhibited different efficiency (for oxidative/reductive chemistry) in photocatalytic H2 evolution. Although the authors provide detailed characterization data of the catalysts, I have some questions about the promotional N effect. The authors should make by consider the following comments.
1.    The N content of NZnO_300 and NZnO_500 must be analyzed and reported; the efficiency differences could be due to both the different N content and N structure here. The authors should clarify this matter.
2.    Which calcination temperature was applied for preparing the ZnO sample?
3.    In the part of XRD characterization, the authors only described the differences between samples, the effect of N on the obtained analysis data here was not discussed like in the other analysis parts.
4.    During the fabrication process, CO2 can be adsorbed and performed the different components among samples. Both the C- and N- make a significant effect on the oxidation state of Zn, so the XPS analysis could be performed to evaluate it. 

Author Response

The authors prepared the N-doped ZnO by the precipitation synthetic method and demonstrated that the thermal treatment during the fabrication process exhibited different efficiency (for oxidative/reductive chemistry) in photocatalytic H2 evolution. Although the authors provide detailed characterization data of the catalysts, I have some questions about the promotional N effect. The authors should make by consider the following comments.
1.    The N content of NZnO_300 and NZnO_500 must be analyzed and reported; the efficiency differences could be due to both the different N content and N structure here. The authors should clarify this matter.

According to the remarks of the referee we performed some very preliminary XPS experiments. Since that measurements it comes out that the amount of Nitrogen is about the 1% and the presence of Carbon is also detected in about 10%. What we could understand from EPR analysis is that we have at least two different types of nitrogen forms, a monomer and a dimer, the monomer is present in both the samples while the dimer is only present in the sample calcined at 300°C.

  1.    Which calcination temperature was applied for preparing the ZnO sample?

The calcination temperature for the preparation of the bare material was 300°C. The information has been added in the experimental session.

  1.    In the part of XRD characterization, the authors only described the differences between samples, the effect of N on the obtained analysis data here was not discussed like in the other analysis parts.

In this study XRD analysis was employed with the aim to verify the purity and crystallinity of the synthetized samples. The relevant information for the work is that the materials shows wurtzitic hexagonal phase, as expected for ZnO sample and no reflection due to remained precursors were recorded. The further information that can be extrapolated is that the doped material calcined at higher temperature (500°C) shows a greater intensity of the XRD reflections revealing larger crystallites as compared to the thermally treated at 300°C. Due to the low N content (and also its small atomic size) was not possible to record any nitrogen structural effect by this technique.

  1.    During the fabrication process, CO2 can be adsorbed and performed the different components among samples. Both the C- and N- make a significant effect on the oxidation state of Zn, so the XPS analysis could be performed to evaluate it. 

See answer to point 1.

Round 2

Reviewer 3 Report

The XPS analysis data must be included with discussion in the manuscript to confirm the N and C content as the authors replied. 

Author Response

The discussion of XPS data has been added in the text of the manuscript. Al the new parts are highlighted in green.